# Humoral Response after Three Doses of mRNA-1273 or BNT162b2 SARS-CoV-2 Vaccines in Hemodialysis Patients

**DOI:** 10.3390/vaccines10040522

**Published:** 2022-03-27

**Authors:** José Jesús Broseta, Diana Rodríguez-Espinosa, Elena Cuadrado, Néstor Rodríguez, José Luis Bedini, Francisco Maduell

**Affiliations:** 1Department of Nephrology and Renal Transplantation, Hospital Clínic of Barcelona, 08036 Barcelona, Spain; dmrodriguez@clinic.cat (D.R.-E.); ecuadrado@clinic.cat (E.C.); fmaduell@clinic.cat (F.M.); 2Diaverum Renal Services Haemodialysis Group, Institut Hemodiàlisi Barcelona, 08036 Barcelona, Spain; nrodriguez@institutocmi.com; 3Department of Biochemistry and Molecular Genetics, Hospital Clínic of Barcelona, 08036 Barcelona, Spain; jlbedini@clinic.cat

**Keywords:** antibody formation, immunogenicity, SARS-CoV-2, COVID-19, COVID-19 vaccines, mRNA vaccines, hemodialysis

## Abstract

The COVID-19 pandemic continues to be a worldwide health issue. Among hemodialysis (HD) patients, two-dose immunization schemes with mRNA vaccines have contributed to preventing severe COVID-19 cases; however, some have not produced a sufficient humoral response, and most have developed a rapid decline in antibody levels over the months following vaccination. This observational, prospective, multi-center study evaluated the humoral response in terms of presence and levels of IgG antibodies to the receptor-binding domain of the S1 spike antigen of SARS-CoV-2 (anti-S1-RBD IgG) to the third dose of SARS-CoV-2 mRNA vaccines, either the mRNA-1273 (Moderna) or BNT162b2 (Pfizer), in 153 patients from three dialysis units affiliated to Hospital Clínic of Barcelona (Spain). Most hemodialysis patients responded intensely to this third vaccine dose, achieving the seroconversion in three out of four non- or weak responders to two doses. Moreover, 96.1% maintained the upper limit or generated higher titers than after the second. BNT162b2 vaccine, active cancer, and immunosuppressive treatment were related to a worse humoral response. Every hemodialysis patient should be administered a third vaccine dose six months after receiving the second one. Despite the lack of data, immunosuppressed patients and those with active cancer may benefit from more frequent vaccine boosters.

## 1. Introduction

The coronavirus disease 2019 (COVID-19) pandemic has been a worldwide health issue, and though cases have risen and fallen during its several waves (or viral variants), it undoubtedly continues to affect us to this day [1]. Some populations have proven to be more prone to present a more severe COVID-19 with more extended hospital admissions and higher mortality rates. This is the case for hemodialysis patients (HD) who not only have worse outcomes but are also at risk of being infected with severe acute respiratory syndrome coronavirus 2 (SARS-CoV-2) due to the higher incidence of outbreaks reported in many dialysis units across several countries. This higher risk is not only because they spend at least 12 hours per week in their hemodialysis units but because they share dressing rooms and ambulances as well [2,3]. In that sense, the different countries’ immunization programs have prioritized hemodialysis patients for the initial vaccination scheme and for receiving booster doses. 

From the many vaccines available for use in the general population, some authors have suggested using the mRNA vaccines for special immune-deficient individuals, such as those receiving hemodialysis, because of their better immunogenicity results [4]. This is not in vain, but due to a widely known immunodeficiency in patients suffering from chronic conditions such as diabetes, oncological, hematological, or autoimmune diseases, chronic kidney disease, old age, or malnutrition [5]. Moreover, there is a weak and waning response in HD patients compared to healthy adults that could be explained by that inadequate immune system response [6]. The immunocompromised uremic state of HD patients can alter the duration and intensity of response to the two-dose SARS-CoV-2 immunization program. This altered immunization response is akin to that seen in response to other immunization schemes such as hepatitis B and influenza virus, where there is often the need for booster doses after completion of the vaccination scheme to allow for these patients to generate and maintain protective antibody levels against the virus [7,8].

Currently, only two mRNA vaccines are approved for clinical use. These are the mRNA-1273 (Moderna) and the BNT162b2 (Pfizer-BioNTech). Among in-center HD patients, current two-dose immunization schemes with any of these mRNA vaccines have contributed to preventing severe COVID-19 cases in most patients up to six months after completing vaccination [9]. Even though two doses of both vaccines have proven to generate potent, though delayed [10], humoral and cellular responses on most HD patients compared to healthy individuals, some of them could not produce a humoral response to vaccination [11]. Moreover, those hemodialysis patients who did respond have presented an increasing loss of detectable IgG antibodies to the receptor-binding domain of the S1 spike antigen of SARS-CoV-2 (anti-S1-RBD IgG) and a rapid decline in antibody levels over the months following vaccination [9,12,13], increasing their risk of SARS-CoV-2 reinfection and severe COVID-19 requiring hospitalization or intensive care [14]. These events have led to the health institutions’ general recommendation that the HD population ought to receive a third booster dose with an mRNA vaccine [9,12,15,16,17].

In light of these data and after the pandemic’s fifth wave, Spain approved administering a third mRNA vaccine dose in HD patients. Since then, several studies have reported a successful experience in the humoral response to a third vaccine dose in most HD patients [15,18,19,20,21]; however, these are single-center, small cohorts where the response to mRNA-1273 vaccination has not been reported. 

This study aims to determine the humoral response in HD patients, especially those who did not respond to the standard two-dose vaccination scheme, and to assess the increase in anti-S1-RBD IgG levels after the third dose of a SARS-CoV-2 mRNA vaccine. 

## 2. Materials and Methods

### 2.1. Study Design and Setting

This is an observational, prospective, multi-center study designed to evaluate the immunogenicity of the third dose of SARS-CoV-2 mRNA vaccines, mRNA-1273 (Moderna, Cambridge, MA, USA) and BNT162b2 (Pfizer-BioNTech, Brooklyn, NY, USA), in HD patients from Hospital Clínic of Barcelona and two affiliated centers, *Centre de Diàlisi i Recerca Aplicada Clínic* and *Institut Hemodiàlisi Barcelona*. Every patient received a complete scheme of three doses with the same vaccine type (i.e., mRNA-1273 or BNT162b2) following the national health authorities’ instruction: high-risk patient profiles such as HD ones who had not been infected in the previous 30 days received a booster. More information regarding the allocation process and other methodological aspects can be found in a previously published study, as this study represents the continuation of the follow-up with a cohort of 201 patients who had already received a two-dose vaccination scheme during February 2020 [11].

### 2.2. Participants

All prevalent adult patients on a maintenance HD program in the three mentioned dialysis facilities were considered for inclusion. Patients were excluded if they had been vaccinated in other healthcare centers, did not receive vaccination because of refusal, were admitted to the hospital during the inclusion period, had a SARS-CoV-2 infection 30 days before the third dose administration, or refused to participate in the study. Included participants were classified according to their previous response status, i.e., non- or weak responders if their anti-S1-RBD IgG levels were inferior to 150 U/mL, and responders if they were equal or superior to this specified amount. 

### 2.3. Humoral Response Assessment

The humoral response to the third dose was measured two weeks after the vaccine administration with the Siemens Healthineers Atellica® IM SARS-CoV-2 IgG (sCOVG) assay, which detects anti-S1-RBD IgG. The assay is considered non-reactive when the result is less than 1 or reactive when greater than or equal to 1, with a maximum measurable range of up to 150 U/mL limiting the antibody response intensity analysis and constituting a potential source of bias. According to the manufacturer, this test has a 96.41% sensitivity (95% CI 92.74–98.54%) and 99.9% specificity (95% CI: 99.63–99.99%).

### 2.4. Other Variables

Other studied variables included were age, gender, dialysis vintage, and comorbidities such as body mass index, diabetes, active cancer, patients with human immunodeficiency (HIV), hepatitis B (HBV) or C (HCV) chronic viral infections, previous SARS-CoV-2 infection, or the use of immunosuppressive therapy. Before initiating the vaccination program, these demographic and medical history data were collected from electronic medical records.

### 2.5. Outcomes

The primary outcome evaluated in this study was the qualitative humoral response to the complete three mRNA vaccine doses. In that sense, seroconversion was used to name those patients whose anti-S1-RBD IgG levels resulted in greater than or equal to 1 in a previous negative person. Secondary evaluated outcomes were quantitative anti-S1-RBD IgG levels among different groups of patients to evaluate the response to the third booster compared to the previous two-dose scheme.

### 2.6. Statistical Methods

Quantitative variables are described as mean and standard deviation, while qualitative variables are reported as absolute and relative frequencies. Univariate analysis was used to estimate the associations between vaccination and outcomes. Differences in qualitative variables were analyzed with the χ2 test or the Fisher’s exact test when one or more expected values were less than five or the data were very unequally distributed among the table’s cells. The normal distribution in the quantitative variables was tested with the Shapiro–Wilk test and Q–Q plots. The quantitative variables’ analysis between the two groups was conducted with the Mann–Whitney U test when non-normal or the independent Student’s t-test when normally distributed. For analyses between more than two groups, related-samples Cochran’s Q test or related-samples Friedman’s two-way analysis of variance by ranks with Bonferroni corrections were performed if variables were qualitative or quantitative, respectively. A two-sided *p*-value inferior to 0.05 was considered statistically significant. Analyses were performed with IBM SPSS® Statistics 26th version (https://www.ibm.com/support/pages/downloading-ibm-spss-statistics-26 accessed on 11 January 2022) and the graphics with GraphPad Prism^®^ 8th version (https://www.graphpad.com/scientific-software/prism/ accessed on 11 January 2022).

### 2.7. Ethical Considerations and Disclosures

All patients provided their written informed consent to participate in this study. The study was conducted following the World Medical Association Declaration of Helsinki, national and local laws, and good clinical practice standards. The institute’s committee on human research approved it. The authors have no conflicts of interest to declare regarding this work. This work has neither received public nor private funding for its implementation.

## 3. Results

### 3.1. Participants

The humoral response to the third dose of mRNA SARS-CoV-2 vaccine was assessed in the 153 included HD patients. Among them, 84 (53.6%) were male and had a median age of 72.12 ± 14.44 years. Regarding the vaccine label allocation, 71 (46.4%) received three doses of BNT162b2 and 82 (53.6%) of mRNA-1273. 

Among the forty-eight patients who were lost during follow-up between the previously mentioned study and this one, twenty died, sixteen received a kidney transplant, four were vaccinated outside the HD center, four refused to receive the third dose, three were transferred to another HD center, and the remaining one was on vacation when blood tests were obtained

### 3.2. Qualitative Response

In total, 149 patients (97.4%) out of the 153 vaccinated individuals seroconverted after the complete three-dose scheme. The clinical characteristics of the four patients (2.6%) who did not respond are seen in Table 1. There were no statistically significant differences in seroconversion between vaccine types, where 68 out of 71 (95.8%) patients who received the BNT162b2 vaccine and 81 out of 82 (98.8%) patients who received the mRNA-1273 (*p* = 0.338) showed a positive anti-S1-RBD IgG test. Percentages of positive serologies to each dose can be seen in Figure 1.

Although there were only 4 non-responders, there were 19 other patients with a weak response, for a total of 23 patients (15% of the cohort). These results are inferior to the 60.8% of non- or weak responders reported after the second dose (*p* < 0.001) and much less than the 88.9% after the first dose (*p* < 0.001), as seen in Figure 2. Predictive non- or weak response factors are shown in Table 2, only finding statistically significant differences for the BNT162b2 vaccine compared to the mRNA-1273 and those under immunosuppressive treatment or with active cancer. 

All non- or weak responders to the third dose were also non- or weak responders to the second dose. However, 70 (75.3%) of these non- or weak responders to the second dose presented a sufficient response to the third one. 

### 3.3. Quantitative Response

Anti-S1-RBD IgG mean levels increased after each mRNA vaccine dose received by this population. At baseline, the measured mean antibody levels were 3.76 ± 18.08 U/mL, which then increased after the first dose to 21.79 ± 47.31, to 97.79 ± 58.88 U/mL after the second, and to 134.35 ± 41.06 U/mL after the booster with the third (Figure 3). All comparisons between vaccine doses were statistically significant (*p* < 0.001). When comparing by vaccine type, although those receiving mRNA-1273 did not have significantly lower anti-S1-RBD IgG levels (1.67 ± 6.33 vs. 6.16 ± 25.55 U/mL, *p* = 0.21) at baseline, after each vaccine dose the differences significantly increased in favor of the mRNA-1273 vaccine (22.83 ± 46.76 vs. 20.58 ± 48.24 U/mL, *p* = 0.002, after the first dose, 107.04 ± 53.76 vs. 74.18 ± 59.97 U/mL, *p* = 0.001, after the second, and 141.01 ± 30.89 vs. 126.67 ± 49.45 U/mL, *p* = 0.016, after the third). 

When comparing antibody levels from the second and third doses, we found that 60 (39.2%) patients maintained the same intensity, with anti-S1-RBD IgG higher than 150 U/mL, 87 (56.9%) had higher levels after the third dose than the second, and there were 6 (3.9%) patients whose levels were lower after the third dose than after the second.

We compared the 20 patients (13.1%) who had a previous COVID-19 infection and found that their anti-S1-RBD IgG levels were higher at baseline (28.68 ± 43.16 vs. 0.01 ± 0.04 U/mL, *p* < 0.001) and after the first two vaccine doses (136.4 ± 33.95 vs. 4.55 ± 11.05, *p* < 0.001, after the first; and 144.93 ± 22.67 vs. 83.8 ± 58.53 U/mL, *p* < 0.001, after the second). However, after the third dose, this difference lost statistical significance (144.42 ± 24.97 vs. 132.84 ± 42.83 U/mL, *p* = 0.18), probably due to a lack of quantification of antibody titers over 150 U/mL.

## 4. Discussion

A third booster dose with an mRNA SARS-CoV-2 vaccine generated an excellent 97.4% rate of positive serologies for anti-S1-RBD IgG in HD patients. Moreover, it significantly reduced the number of HD patients unable to mount a humoral response, or who only had generated a weak response after the recommended standard two-dose vaccination scheme from 60.8% to 15%. Even though our work has reported that 23 patients still were unable to produce an adequate humoral response after three vaccination doses, it means that this booster has enabled every three out of four individuals to achieve an optimal humoral response. Indeed, only four patients did not respond at all, though they had striking clinical characteristics or medical history that may justify this hampered immune response. Currently, there are few and variable data on the effect of a third booster vaccine dose, and most only evaluate the BNT162b2 vaccine response on those HD patients who have previously had a suboptimal response [15,18,20,21,22]. The results found in in our work are almost identical to those reported by Bensouna et al. [18] and are similar to other published data [20,22].

Ensuring a long-lasting immune response in HD patients to SARS-CoV-2 is a crucial public health objective. As it has been thoroughly reported in various published works, SARS-CoV-2 infection is highly contagious and has been the culprit of disease outbreaks in countless HD centers [23,24]. However, even though lower infection rates remain one of the main healthcare outcomes, we acknowledge that the most disturbing data found during COVID-19’s first wave was not the proportion of infected individuals, but the extremely high mortality rate seen in this population, with around one in every three patients dying from this disease [3]. Therefore, the true triumph of vaccination lies in obtaining a reduction in significant clinical outcomes such as fewer and shorter hospital admissions, as well as less severe cases with fewer ICU admissions and lower mortality associated with COVID-19. We now know that, at least in HD patients, SARS-CoV-2 vaccination with two doses of either mRNA1273 or BNT162b2 is highly protective of progression to severe COVID-19 at least within six months of being immunized [25].

It is in this sense that the waning response reported by other groups [16] and ours [9] calls for the administration of additional booster doses in the HD population. Compared to other cohorts [20,22] that received three doses and our previous findings after two doses in this same population, we found that immunosuppressive therapy remains a significant factor associated with a poor response [11]. A non-negligible number of hemodialysis patients are on immunosuppressive medication as treatment for autoimmune diseases such as systemic vasculitis, systemic lupus erythematosus, hematologic malignancies [26], and most notably as treatment for a failed kidney transplant or another functioning solid organ transplant [27]. It is no surprise to find immunosuppressive therapy as a risk factor, particularly given the humoral response to COVID-19 vaccines reported in vaccinated immunodeficient kidney transplant recipients. A recent study by Stumpf et al. has shown a low seroconversion rate of around 30% after two doses of the BNT162b2 vaccine that rose to 55% after three doses [28]. The low seroconversion seen in this population is associated with a higher incidence of severe COVID-19 and greater mortality [25].

Compared to immunosuppressive medication, which remained significantly associated with a worse response after both two and three doses, the previously associated factors with two doses lost significance after the third one. For instance, patients with a past history of COVID-19 who had already received an immunological stimulus after their natural viral exposure had previously presented higher antibody levels after the first and second mRNA vaccine doses. In contrast, after the third dose, most COVID-19 naïve HD patients produced anti-S1-RBD IgG levels as high as those who had been previously infected. This could mean that the three cumulative vaccine doses can elicit a humoral response as potent as natural immunity or that these patients’ innate immunity had waned with the passing months, as has been reported in the general population [29].

In addition, this work is the first to report data on the humoral response to a third dose of the mRNA-1273 vaccine and, even though this study was not designed to determine a difference in effectiveness between these two vaccine brands, we found that those who received the mRNA-1273 vaccine were not only more likely to have detectable anti-S1-RBD IgG (98.8%), but also to have a lower proportion of non- or weak responders. However, these results must be regarded with caution, as in data from previously published articles, we found that these differences disappeared at least 3 and 6 months after the second dose of vaccine [9,13].

Regarding other immunosuppressed populations, a recently published study evaluated the humoral response in hemato-oncological patients to two doses of the BNT162b2 vaccine [30]. This cohort included patients with a myriad of oncological diseases ranging from lung and colon cancer to multiple myeloma and myelodysplastic syndrome who received various chemotherapeutic treatments. Hematological patients had lower antibody titers and an increased ratio of seroreversion than those with solid tumors [31]. Though this difference could be due to the use of B cell-targeting agents, both solid organ and hematologic cancers who received three doses of the vaccine had a slightly lower proportion of positive anti-S1-RBD IgG than HD patients (94% and 88%, respectively). Similarly, patients with multiple sclerosis on treatment with anti-CD20 agents had a strikingly weak response to a third dose of the BNT162b2 vaccine, where only one out of sixteen patients developed clinically relevant antibody titers [32]. Even though there is no previously reported association between active cancer and a null or weak response in HD patients, we did find this association in our studied population. We hypothesize that this may be related to the cytotoxic chemotherapy that these patients were receiving. Two of them were receiving cyclophosphamide treatment for dysproteinemic cancer, while the remaining one received a platinum-based agent as the treatment for colon cancer. The cytotoxic chemotherapeutic therapies operate in a similar fashion to immunosuppressive medication, acting, in part, by depleting lymphocytes, therefore reducing the patients’ immune system’s capacity to be stimulated to produce antibodies as a response to vaccination [33]. 

Another immunodeficient population is elderly individuals; however, there are scarce data regarding three-dose studies from this group. Research on their response to two doses has found age to be unrelated to the intensity of the humoral response, though an association was found with diabetes mellitus, cancer, and previous SARS-CoV-2 infection [34]. This low serological response seen in elders with type 2 diabetes has not been observed in the diabetic population as a whole, where no differences were reported between them, and the healthy individuals used as the control group [35]. In our cohort, age was inversely related to anti-S1-RBD IgG levels after the standard two-dose vaccination; nevertheless, this difference disappeared after the administration of three doses. Similarly, we found no difference in seroconversion rates between diabetic HD patients and non-diabetics. 

Our study has some limitations. Firstly, our laboratory was only able to report anti-S1-RBD IgG levels up to 150 U/mL. This could mean that a difference may exist on higher ranges that we have not measured, which would be in accordance with that reported in other works [18]. Secondly, there is still a high heterogenicity of antibody measurement assays that make our antibody levels not comparable to that of every other published paper. Moreover, these tests were approved by the healthcare authorities in the emergency setting and still need to be validated and approved. Thirdly, a control group was not available to make comparisons, as we considered that it would be unethical to deprive a group of patients of the booster dose. 

## 5. Conclusions

Most hemodialysis patients responded intensely to this third vaccine dose, achieving the seroconversion in three out of four non- or weak responders to two doses. Moreover, 96.1% maintained the upper limit or generated higher titers than after the second one. Therefore, we recommend every hemodialysis patient receives a third vaccine dose six months after receiving the second one. Further studies are needed to evaluate the maintenance of these immune responses and the potential need for more booster doses in every HD patient or specific subpopulations. Promising data on a fourth dose in kidney transplant recipients are being gathered, and a fifth one is being scheduled, as they have a poor response and worse outcomes than their peers who remain on dialysis.

## Figures and Tables

**Figure 1 vaccines-10-00522-f001:**
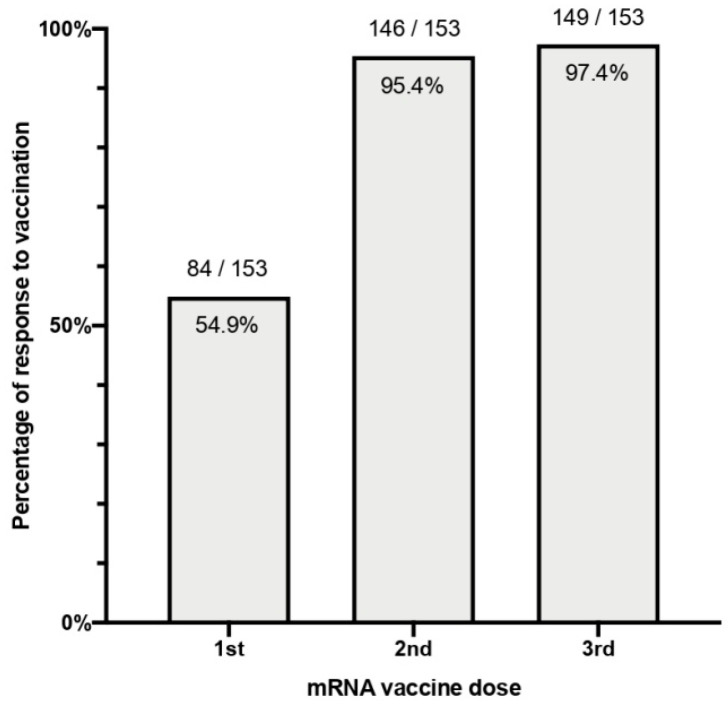
Percentage of patients with humoral response to each dose of any SARS-CoV-2 mRNA vaccine.

**Figure 2 vaccines-10-00522-f002:**
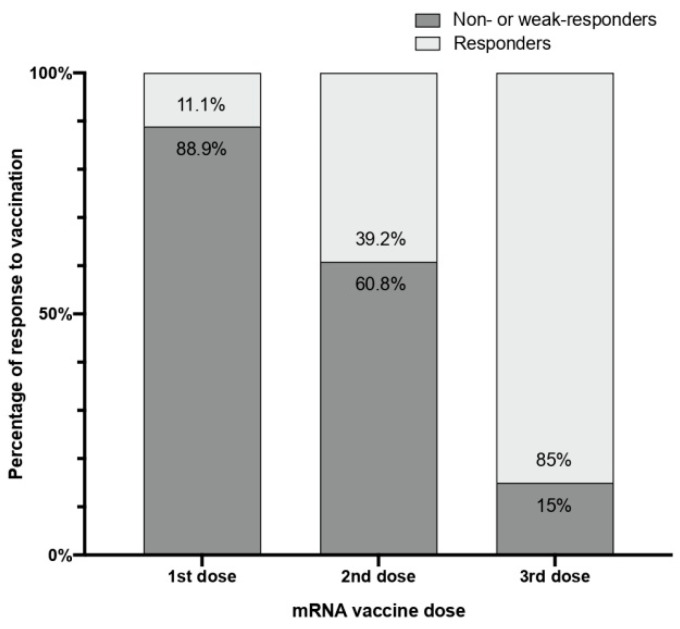
Percentages of non- or weak responders (anti-S1 RBG IgG levels < 150 U/mL) and responders (anti-S1 RBG IgG levels ≥ 150 U/mL) to each dose of any SARS-CoV-2 mRNA vaccine.

**Figure 3 vaccines-10-00522-f003:**
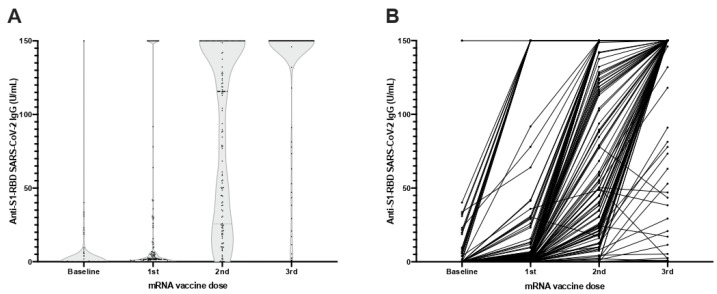
Anti-S1-RBD IgG levels at baseline and after each dose with any SARS-CoV-2 mRNA vaccine (**A**) Violin plots of anti-S1-RBD IgG levels. (**B**) Before–after plot of anti-S1-RBD IgG levels.

**Table 1 vaccines-10-00522-t001:** Clinical characteristics of non-responders to three doses of SARS-CoV-2 mRNA vaccination.

N	Sex	Age	Vaccine	ESKD Cause	Dialysis Vintage (Months)	Immunosuppressive Treatment	Comorbidities	Previous Humoral Response
1	Male	88	BNT162b2	CNI toxicity	58	Tacrolimus	Liver Transplant	No response after two doses
2	Male	65	BNT162b2	DKD	40	No	POEMS syndrome Obesity, Diabetes	No response after two doses
3	Female	82	BNT162b2	Unknown	32	Cyclophosphamide	Multiple myeloma	Response lost 3 months after the two doses
4	Male	54	mRNA-1273	Unknown	114	Tacrolimus and Mycophenolic acid	Liver transplant Colon cancer HIV, HBV	No response after two doses

ESKD: end-stage kidney disease; CNI: calcineurin inhibitors; DKD: Diabetic kidney disease; POEMS: polyneuropathy, organomegaly, endocrinopathy, monoclonal protein, skin changes; HIV: human immunodeficiency virus; HBV: hepatitis B virus.

**Table 2 vaccines-10-00522-t002:** Demographic and clinical characteristic comparisons between non- or weak responders and responders to the third SARS-CoV-2 mRNA vaccine dose.

Variable	Total *n* = 153	Non- or Weak Responders (<150 U/mL) *n* = 23 (15%)	Responders (≥150 U/mL) *n* = 130 (85%)	*p*-Value	Odds Ratio ^1^ (95% Confidence Interval)
Age > 75 years	74 (48.4%)	12 (52.2%)	62 (47.7%)	0.69	1.2 (0.49–2.9)
Male sex	83 (54.2%	12 (52.2%)	71 (54.6%)	0.83	1.1 (0.45–2.68)
Dialysis vintage (over the 50th percentile)	75 (49%)	13 (56.5%)	62 (47.7%)	0.44	1.43 (0.58–3.48)
BNT162b2 vaccine	71 (46.4%)	16 (69.6%)	55 (42.3%)	0.02	3.12 (1.2–8.1)
Overweight	45 (29.4%)	7 (30.4%)	38 (29.2%)	0.97	1.06 (0.4–2.7)
Obesity	27 (17.6%)	3 (13%)	24 (18.5%)	0.53	0.66 (0.18–2.4)
Diabetes	55 (36.2%)	7 (30.4%)	48 (37.2%)	0.53	0.74 (0.28–1.92)
Immunosuppressive therapy	11 (7.2%)	5 (21.7%)	6 (4.6%)	0.01	5.74 (1.59–20.83)
Active cancer	5 (3.3%)	3 (13%)	2 (1.6%)	0.03	9.52 (1.5–58.82)
HIV chronic infection	4 (2.6%)	1 (4.3%)	3 (2.3%)	0.49	1.9 (0.19–19.23)
HBV chronic infection	6 (3.9%)	1 (4.3%)	5 (3.9%)	1	1.13 (0.13–10.1)
Previous SARS-CoV-2 infection	20 (13.1%)	1 (4.3%)	19 (14.6%)	0.18	3.77 (0.48–29.6)

^1^ Odds ratios are calculated to indicate the risk estimate to become non- or weak responder.

## Data Availability

The data that support the findings of this study are available on request from the corresponding author, José Jesús Broseta.

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
