# Peer review of "Humoral Response after Three Doses of mRNA-1273 or BNT162b2 SARS-CoV-2 Vaccines in Hemodialysis Patients"

_vaccines, 2022, doi:10.3390/vaccines10040522_

Round 1

Reviewer 1 Report

The paper “Humoral response after three doses of mRNA-1273 or BNT162b2 SARS-CoV-2 vaccines in hemodialysis patients”   evaluating the antibody response of a cohort of hemodialysis patients to the administration of a booster dose of the two SRS-CoV-2 RNA vaccines has an interesting originality.

In indeed, although it was a single-center study, it included an adequate number of subjects so that it could provide results applicable on a larger scale.

The study appears to be well organized , with adequate presentation of findings and a fairly complete discussion.

Concerning the specific contents, it can be considered a regret that the authors did not consider the possibility of including in this survey patients with treatments other than Hemodialysis such as Peritoneal Dialysis, or Kidney Transplantation. This might have verified if a different modality of treatment of Uremia is able to modify the immune response to vaccines. However, the submitted document may be reasonably accepted for publication.

Some questions can be addressed to the authors for some clarifications or suggestions for discussion.

  • The selected cohort of patients, was it part of a larger group of patients some of whom were already infected with Covid-19?
  • If so, specifically how many had contracted Covid-19 and recovered and how many had died?
  • When the study was conducted, which mutation of the SARS-CoV-2 virus was present in the country (native, delta, omicron)?
  • Have any patients been infected after the second dose of vaccine?
  • Did they receive a third dose or were they considered to be already reinforced in antibody stimulation?
  • The authors highlight the progressive increase of the antibody titer after the third dose, even in those patients who seemed refractory after the 1st vaccine cycle, how do they explain this response that seems to match that of the general population?  
  • Based on the experience from this study, the authors believe that the immune response may still improve after a 4th dose, as it seems many nations are inclined to do in immunocompromised patients such as hemodialysis?

Author Response

We really appreciate the reviewer's comments. We agree that including peritoneal dialysis patients and kidney recipients would be of interest; however, the healthcare system's different prioritization of the third dose vaccination made this impossible. 

Regarding the arisen questions and considerations, we answer them independently:

The selected cohort of patients, was it part of a larger group of patients some of whom were already infected with Covid-19? If so, specifically how many had contracted Covid-19 and recovered and how many had died?

Thank you very much for your comments. We included all prevalent patients on maintenance HD program in the dialysis facilities that depend on our center. That includes patients previously infected with SARS-CoV-2 and, thus, with basal antibodies (as shown in figure 3). 

As said in the 3.1 section: This study represents the continuation of the follow-up of a cohort of 201 patients who had already received a two-dose vaccination scheme. Forty-eight patients were lost during follow-up. Twenty died, 16 received a kidney transplant, four were vaccinated outside the center, four refused to receive the third dose, three were transferred to another hemodialysis center, and the remaining one was on vacation when blood tests were obtained.

However, no dead patients before the third dose administration were included as data on the third dose would not have been available. In the 3.3 section, we added the number of previously infected patients to clarify this point: When comparing the 20 patients (13.1%) that had passed a COVID-19 infection, anti-S1-RBD Ig G levels were higher at baseline

When the study was conducted, which mutation of the SARS-CoV-2 virus was present in the country (native, delta, omicron)?

Although we do not have information on the exact variant of the SARS-CoV-2, and only some tests are performed the genetic study as a sentinel surveillance, comething we believe is not sufficient to be written in the main text, at these days the predominant variant in our region was the delta one, although we were in the gap between the fifth and the sixth omicron wave. 

Have any patients been infected after the second dose of vaccine? Did they receive a third dose or were they considered to be already reinforced in antibody stimulation?

Yes, they did. This data, however, has been previously published in Rodríguez-Espinosa D, Montagud-Marrahi E, Cacho J, et al. Incidence of severe breakthrough SARS-CoV-2 infections in vaccinated kidney transplant and haemodialysis patients [published online ahead of print, 2022 Feb 21]. J Nephrol. 2022;1-10. doi:10.1007/s40620-022-01257-5

And they did receive a third dose if the infection had occurred at least four weeks before the positive test or symptoms initiation. 

The authors highlight the progressive increase of the antibody titer after the third dose, even in those patients who seemed refractory after the 1st vaccine cycle, how do they explain this response that seems to match that of the general population? Based on the experience from this study, the authors believe that the immune response may still improve after a 4th dose, as it seems many nations are inclined to do in immunocompromised patients such as hemodialysis?

As the reviewer will perfectly know, immune memory is boosted by each new dose and, thus, booster doses maximize the protection. Moreover, for vaccines that do not confer lifelong immunity, booster doses give a further round of exposure to refresh this immune memory. There is still a lack of data to state the need or not of a 4th dose firmly. It seems clear that hemodialysis patients' response to vaccines in terms of the immune response, but also hard outcomes, is closer but not equal to that of the general population than their kidney transplant recipient peers as it takes more time for them to become seropositive and their antibody levels drop significantly more rapidly: Therefore and even though there is no compelling data backed by a randomized control trial, it seems that this population will benefit from booster COVID-19 vaccines, especially given the high mortality seen in haemodialysis patients during the pre-vaccination era compared to post-vaccination (Rodríguez-Espinosa, D., Montagud-Marrahi, E., Cacho, J. et al. Incidence of severe breakthrough SARS-CoV-2 infections in vaccinated kidney transplant and haemodialysis patients. J Nephrol (2022). https://doi.org/10.1007/s40620-022-01257-5).

Reviewer 2 Report

The following points should be addressed:

  1.  The mentioning of reference #7 in line 40 is misleading. The authors use this reference to highlight the risk of reinfection/severe disease due to dropping antibody titers following vaccination. This case report by the same authors describes a HD patient who died of COVID-19. However, there is no vaccination record for this patient. I would remove this.
  2. As a general question and relating to the sentence in line 40 about "weak and waning response compared to otherwise healthy adults", has the same antibody test the authors use in all their studies been used for any studies in healthy individuals? If not, how can they justify this statement?
  3. Please double-check all mean values provided in section 3.3 Quantitative response as they do not match the means shown in Figure 3A. 
  4. Sentence starting in line 178: It's a bit confusing that the authors first talk about not significant differences in anti-S1-RBD titers (it should be included that this is baseline) between the vaccine types, and then say in the second part of the sentence that the difference reversed. A difference that is not there cannot reverse. Please rephrase.
  5. In line 193 the authors state that the difference in antibody titers between Moderna and Pfizer lost the statistical significance. It should be included here or at the very least in the discussion that this lack of difference is most likely due to the antibody titers hitting the ceiling/upper limit of the assay.
  6. In general, I am not sure how well the employed statistics represent the data as antibody values were capped at 150U/ml. I know this is an inherent limitation of the study.

Author Response

The mentioning of reference #7 in line 40 is misleading. The authors use this reference to highlight the risk of reinfection/severe disease due to dropping antibody titers following vaccination. This case report by the same authors describes a HD patient who died of COVID-19. However, there is no vaccination record for this patient. I would remove this.

Thank you for your comment. You are completely right. We have modified the reference by another that does reinforce the idea we wrote. 

As a general question and relating to the sentence in line 40 about "weak and waning response compared to otherwise healthy adults", has the same antibody test the authors use in all their studies been used for any studies in healthy individuals? If not, how can they justify this statement?

Although we have had no control group in our studies, the Siemens Atellica test has been performed in different works with healthcare professionals, patients, and healthy adults. We give some examples of works in which it has been used: 

Conklin J, Arroyo J, Kumar R, Patibandla S. Two SARS-CoV-2 serology assays detect antibodies in the sera of individuals diagnosed with SARS-CoV-2 Omicron variant [published online ahead of print, 2022 Feb 24]. Clin Biochem. 2022;S0009-9120(22)00058-3. doi:10.1016/j.clinbiochem.2022.02.010

Freeman J, Conklin J. Standardization of two SARS-CoV-2 serology assays to the WHO 20/136 human standard reference material. J Virol Methods. 2022;300:114430. doi:10.1016/j.jviromet.2021.114430

Epaulard O, Buisson M, Nemoz B, et al. Persistence at one year of neutralizing antibodies after SARS-CoV-2 infection: Influence of initial severity and steroid use. J Infect. 2022;84(3):418-467. doi:10.1016/j.jinf.2021.10.009

Cinislioglu AE, Demirdogen SO, Cinislioglu N, et al. Variation of Serum PSA Levels in COVID-19 Infected Male Patients with Benign Prostatic Hyperplasia (BPH): A Prospective Cohort Studys. Urology. 2022;159:16-21. doi:10.1016/j.urology.2021.09.016

Carta M, Marinello I, Cappelletti A, et al. Comparison of Anti-SARS-CoV-2 S1 Receptor-Binding Domain Antibody Immunoassays in Health Care Workers Before and After the BNT162b2 mRNA Vaccine. Am J Clin Pathol. 2022;157(2):212-218. doi:10.1093/ajcp/aqab107

Lee N, Jeong S, Park MJ, Song W. Comparison of three serological chemiluminescence immunoassays for SARS-CoV-2, and clinical significance of antibody index with disease severity. PLoS One. 2021;16(6):e0253889. Published 2021 Jun 29. doi:10.1371/journal.pone.0253889

Florin L, Maelegheer K, Vandewal W, Bernard D, Robbrecht J. Performance Evaluation of the Siemens SARS-CoV-2 Total Antibody and IgG Antibody Test. Lab Med. 2021;52(6):e147-e153. doi:10.1093/labmed/lmab027

Ward MD, Mullins KE, Pickett E, et al. Performance of 4 Automated SARS-CoV-2 Serology Assay Platforms in a Large Cohort Including Susceptible COVID-19-Negative and COVID-19-Positive Patients. J Appl Lab Med. 2021;6(4):942-952. doi:10.1093/jalm/jfab014

Please double-check all mean values provided in section 3.3 Quantitative response as they do not match the means shown in Figure 3A. 

Thank you for your comments. Note that violin plots are constructed with medians and not means. 

Sentence starting in line 178: It's a bit confusing that the authors first talk about not significant differences in anti-S1-RBD titers (it should be included that this is baseline) between the vaccine types, and then say in the second part of the sentence that the difference reversed. A difference that is not there cannot reverse. Please rephrase.

Thank you for your comment. After a new reading, we agree this sentence is really confusing. We have rewritten it as:  "When comparing by vaccine type, although those receiving mRNA-1273 had not significant lower anti-S1-RBD IgG levels (1.67 ± 6.33 vs. 6.16 ± 25.55 U/mL, p = 0.21) at baseline, after each vaccine dose, the differences significantly increased in favour of mRNA-1273 (22.83 ± 46.76 vs. 20.58 ± 48.24, p = 0.002, after the first dose, 107.04 ± 53.76 vs. 74.18 ± 59.97, p = 0.001, after the second, and 141.01 ± 30.89 vs. 126.67 ± 49.45, p = 0.016, after the third)"

In line 193 the authors state that the difference in antibody titers between Moderna and Pfizer lost the statistical significance. It should be included here or at the very least in the discussion that this lack of difference is most likely due to the antibody titers hitting the ceiling/upper limit of the assay.

Thank you for your comment. We agree with the author that this can be a cause, although we also think that this limitation was addressed with this sentence "Our study has some limitations. Firstly, our laboratory was only able to report anti-S1-RBD IgG levels up to 150 U/mL, which could mean that a difference may exist on higher ranges that we have not measured, which would be in accordance with that reported in other works" We have clarified, though, the writing of the sentence.

In general, I am not sure how well the employed statistics represent the data as antibody values were capped at 150U/ml. I know this is an inherent limitation of the study.

We agree this is an important limitation, but it was a challenging time for our microbiology department and they were not able to do dilutions. A median comparison like Cohen's kappa coefficient may have been another way to compare these results, but it has its own controversies, and we have previously published papers with the same statistical tests performed in this one.

Reviewer 3 Report

See comments in attached pdf file

Author Response

Thank you for your comments.

We answer your concerns in the following lines:

1.- Rephrased.

2.- Rephrased.

3.- We politely disagree with the reviewer, as this is the correct referencing method.

4.- Corrected.

5.- More data avialable in the referenced paper.

6.- This is a arbitrary treshold, as there is a lack of evidence. However, we do know that this titers correlate with antiviral neutralization thanks to .

7.- As previously said, "the assay is considered non-reactive when the result is 80
less than 1 or reactive when greater than or equal to 1", and that is the value for wich specifity and sensitivity are calculated.

8.- Yes, the testing scheme was that recommended by the manufacturer to achieve the maximal humoral response.

9.- Seroconversion is defined as a anti-S1 IgG result superior to 1 in a previous negative person. It has been added a clarification of the term in methods (section 2.5).

10.- As no statistical significance was achieved, we prefered to group both results. The message of this work is that any mRNA vaccine performs well as we do not really know if this difference on humoral antibodies predicts better hard outcomes.

11.- Modified.

12.- Corrected.

13.- Modified.